# Learning Dispersed Embeddings on Hyperspheres

## Abstract

Learning well-separated features in high-dimensional spaces, such as text or image *embeddings*, is crucial for many machine learning applications. Achieving such separation can be effectively accomplished through the *dispersion* of embeddings, where unrelated vectors are pushed apart as much as possible. By constraining features to be on a *hypersphere*, we can connect dispersion to well-studied problems in mathematics and physics, where optimal solutions are known for limited low-dimensional cases. However, in representation learning we typically deal with a large number of features in high-dimensional space, which makes leveraging existing theoretical and numerical solutions impossible. Therefore, we rely on gradient-based methods to approximate the optimal dispersion on a hypersphere. In this work, we first give an overview of existing methods from disconnected literature. Next, we propose new reinterpretations of known methods, namely Maximum Mean Discrepancy (MMD) and Lloyd's relaxation algorithm. Finally, we derive a novel dispersion method that directly exploits properties of the hypersphere. Our experiments show the importance of dispersion in image classification and natural language processing tasks, and how algorithms exhibit different trade-offs in different regimes.

## 1 Introduction

Dispersion[1] of embeddings encourages spreading out a large amount of high-dimensional embedding vectors on the surface of the $d$-dimensional unit hypersphere (Liu et al., 2021). Clustering of the embeddings, *i.e.*, occurrence of semantically distant embeddings that are close to each other in terms of distance metric, is a known problem, and it has been shown before that it negatively impacts the performance of the downstream tasks, such as image classification (Wang & Isola, 2020; Liu et al., 2021; Trosten et al., 2023), image generation (Liu et al., 2021), text classification (Wang & Isola, 2020) and text generation (Tokarchuk & Niculae, 2024). Mettes et al. (2019) also argue that directly minimized maximum similarity of the points on the hypersphere is superior to uniformly obtained samples (Hicks & Wheeling, 1959; Muller, 1959), since it explicitly encourages separation between points.

In general, the problem of spreading $N$ points on the surface of $d$ dimensional sphere, such that the angular distance between any two points is maximal, is an open mathematical problem known as the Tammes problem (Tammes, 1930). The optimal solutions for this problem are known for small values of $d$ and $N$ (Fejes, 1943; Danzer, 1986; Waerden van der & Schütte, 1951; Robinson, 1961; Musin & Tarasov, 2012; 2015). The Tammes problem can also be formulated as a problem of finding a spherical code (Conway et al., 1999) with minimal cosine similarity value for given $d$ and $N$ (Cohn, 2024). However, we typically deal with a large number of dimensions and many points when learning, e.g., text embeddings for ML tasks. Thus, we can rely on gradient optimization methods to approximate the optimal configuration on the hypersphere. Dispersion is also closely connected to the contrastive learning (Chen et al., 2020a; He et al., 2020; Hjelm et al., 2019; Chen et al., 2020b), where model outputs corresponding to different classes are pushed away from each other. Wang & Isola (2020) in particularly showed that widely used contrastive learning objective can be interpreted in terms of "alignment" (similar features for similar samples) and "uniformity"

---

[1]In the literature, the term "uniformity" is also used. However, to highlight the difference with samples from the uniform distribution, we use "dispersion" instead.

(feature distribution is close to uniform distribution). In our work we focus on parameter dispersion, which can more easily be quantified. We study several dispersion objectives in order to find an approximate solution to the dispersion problem on the unit hypersphere. In particular, we reinterpret Maximum Mean Discrepancy (MMD, Gretton et al., 2012) as a method for dispersing an arbitrary number of high-dimensional points, adapt Lloyd's algorithm (Lloyd, 1982), and propose sliced dispersion that directly exploits properties of the hypersphere. We compare them to the previously proposed methods based on pairwise distances (Mettes et al., 2019; Sablayrolles et al., 2019; Liu et al., 2021; 2018b; Wang & Isola, 2020). We showcase the performance of those objectives by approximating optimal Tammes problem solutions and learning dispersed representation both for computer vision and natural language processing tasks. Our results show that there is a dependence between task performance and respective dispersion of the features. Additionally, we highlight that using Riemannian optimization (Bonnabel, 2013; Becigneul & Ganea, 2019) on the hypersphere, rather than projecting parameters to the sphere at each gradient update, benefits dispersion and overall task performance.

Our contributions are the following:

- Review connections between several proposed dispersion regularizers based on pairwise distances, and give a new interpretation and motivation in terms of maximum mean discrepancy (MMD);
- Propose two new methods for approximating optimal dispersion (Lloyd and Sliced);
- Provide empirical comparison among dispersion optimization methods on tasks from vision and language processing;
- Investigate the impact of Riemannian optimization for dispersion.

Moreover, our implementation and experiment code will be released as an open-source library upon publication.

## 2 DISPERSION ON THE HYPERSPHERE

First we discuss the notation we are going to use throughout the paper, give the definition of "dispersion" and review existing approximate methods to estimate optimal dispersion.

### 2.1 NOTATION AND BACKGROUND

We denote by $\mathbb{S}_d$ the $d$-dimensional hypersphere embedded in $\mathbb{R}^{d+1}$, *i.e.*, $\mathbb{S}_d = \{x \in \mathbb{R}^{d+1} \mid \|x\| = 1\}$. For $u, v \in \mathbb{R}^{d+1}$ we denote their Euclidean inner product by $\langle u, v \rangle := \sum_{i=1}^{d+1} u_i v_i$. The hypersphere is an embedded Riemannian submanifold of $\mathbb{R}^{d+1}$. The tangent space of the sphere at a point $x$ is $T_x \mathbb{S}_d := \{v \in \mathbb{R}^{d+1} \mid \langle x, v \rangle = 0\} \simeq \mathbb{R}^d$, and the Riemannian inner product on it is inherited from $\mathbb{R}^{d+1}$, *i.e.*, for $u, v \in T_x \mathbb{S}_d$, $\langle u, v \rangle_x := \langle u, v \rangle$. The geodesic distance on a hypersphere is $d(x, x') = \cos^{-1}(\langle x, x' \rangle)$. As a special case, for $d = 1$ it is more convenient to work in an isomorphic angular parametrization, *i.e.*, $\mathbb{S}_1 \simeq \{\theta \mid -\pi \leq \theta < \pi\}$ with $d(\theta, \theta') = |\theta - \theta'|$: the embedding of $\mathbb{S}_1$ into $\mathbb{R}^2$ is given by $\theta \to (\cos\theta, \sin\theta)$. We reserve the use of Greek letters $\tau, \theta, \phi$ for 1-d angles. We denote by $\Pi_n$ the set of permutations of $(1, \ldots, n)$.

We use roman capitals, *i.e.*, $X = (x_1, \ldots, x_n)$, to denote an (ordered) collection, or configuration, of $n$ points on the same sphere, *i.e.*, each $x_i \in \mathbb{S}_d$. We use sans-serif capitals, *i.e.*, Y, to denote a random variable.

### 2.2 MEASURES OF DISPERSION

To measure the dispersion of the set of embeddings $X$ on the unit hypersphere, we consider two different metrics.

**Minimum distance.** Dispersion requires that no two points be too close, so following Zhou et al. (2022) we employ a minimum distance metric:

$$d_{\min}(X) = \min_{x_i, x_j \in X, i \neq j} d(x_i, x_j), \tag{1}$$

where $d(x_i, x_j)$ is the geodesic distance from §2.1.

**Spherical variance.** Spherical variance (Jammalamadaka & Sengupta, 2001; Mardia, 1975) originates from directional statistics and is defined for finite $X \subseteq \mathbb{S}_d$ as

$$\mathrm{svar}(X) = 1 - \overline{R}, \text{ where } \overline{R} = 1/n \sum_i x_i. \tag{2}$$

Spherical variance is a key quantity in the Raleigh test for uniformity on the hypersphere $\mathbb{S}_d$ (Mardia & Jupp, 1999, p. 206–208), which uses $(d+1)n\overline{R}^2$ as test statistic.

The presented dispersion measures offer complementary perspectives of the dispersion of the embeddings, but are insufficient when considered in isolation. The minimum distance only depends on the two closest embeddings: embeddings can be spread out in a near perfect configuration, whilst having a minimum distance close to zero. Similarly, large spherical variance does not imply well dispersed embeddings (consider embeddings clustered around two antipodes.) In addition, neither method is well-suited for gradient optimization. The gradient of $d_{\min}$ depends only on the closest pair of points and would lead to impractically slow algorithms. As for spherical variance, since the Euclidean gradient of $\overline{R}$ is orthogonal to the surface of the hypersphere $\mathbb{S}_d$, its Riemannian gradient is null. Similarly to spherical variance, the Raleigh test cannot be used as minimization objective to disperse embeddings.

There are many other ways to measure dispersion (Marbut et al., 2023), but in the scope of this work we focus on two described above simple metrics.

## 2.3 PAIRWISE MEASURES FOR DISPERSION

Using pairwise distances for dispersion on the hypersphere has been long of interest for the machine learning community (Sablayrolles et al., 2019; Mettes et al., 2019; Wang et al., 2020b; Trosten et al., 2023; Liu et al., 2018b; 2021). All these works use pairwise-based as a backbone for their objectives, which leads to quadratic complexity and require calculating a matrix of pairwise distances.

**Max-Min Distance.** To achieve better dispersion on hypersphere, a variety of works focus on maximizing minimum distance (or equivalently minimizing maximum pairwise similarity) (Mettes et al., 2019; Wang et al., 2020b; Liu et al., 2021). In this case, for each embedding vector, only its nearest neighbor and the embedding itself are updated. The regularizer takes the form:

$$\mathcal{L}_{\text{Max-Min}} = -\frac{1}{n} \sum_{i=1}^{n} \min_{j \neq i} d(x_i, x_j), \tag{3}$$

where $d$ can be the cosine distance (MMCS, Mettes et al., 2019), the geodesic distance (MMA, Wang et al., 2020b), or the euclidean distance (Liu et al., 2021).

**Differential Entropy Dispersion.** Using maximum entropy regularization is a known technique in machine learning (Meister et al., 2020; Ahmed et al., 2019; Pereyra et al., 2017; Liu et al., 2018a) aiming to encourage diversity of the output and improve generalization, *i.e.*, higher entropy pushes the output distribution closer to the uniform distribution. Sablayrolles et al. (2019) proposed to extend this idea for the continuous space, and directly maximize differential entropy on hypersphere. To this end, they propose to use Kozachenko-Leonenko estimator (Leonenko, 1987)

$$\mathcal{L}_{\text{KoLeo}} = -\frac{1}{n} \sum_{1}^{n} \log \min_{i \neq j} \|x_i - x_j\|. \tag{4}$$

Note that the following bound holds between $\mathcal{L}_{\text{KoLeo}}$ and the logarithm of the max-min distance:

$$-\frac{1}{n} \log \Big( \sum_{i=1}^{n} \min_{j \neq i} d(x_i, x_j) \Big) \geq -\frac{1}{n} \sum_{1}^{n} \log \min_{i \neq j} \|x_i - x_j\|.$$

**MHE.** Inspired by Thomson problem (Gautam & Vaintrob, 2013), Liu et al. (2018b; 2021) proposed to use *minimum hyperspherical energy* (MHE) in order to ensure separation of the points on hypersphere.

$$\mathcal{L}_{\text{MHE}} = \sum_{i=1}^{n} \sum_{j=1, j \neq i}^{n} f_s(\|x_i - x_j\|), \tag{5}$$

where $f_s(\cdot)$ is a decreasing real-valued function and $\| \cdot \|$ is an Euclidean distance. Liu et al. (2018b; 2021); Lin et al. (2020) used $f_s(z) = z^{-s}$, $s > 0$, known as Riesz s-kernel:

$$k_s(x_i, x_j) = \begin{cases} d(x_i, x_j)^{-s}, \ s > 0, \\ \log\big(d(x_i, x_j)^{-1}\big), s = 0 \end{cases}$$

where $d$ can be Euclidean or geodesic distance. Riesz s-energy has many applications in various mathematical and physics problems, and connects to the Gaussian kernel through the Laplace transformation (Borodachov et al., 2019).

**Uniformity.** Wang & Isola (2020) introduced the *uniformity* measure for representation learning based on pairwise Gaussian potential:

$$\mathcal{L}_{\text{uniform}} = \log \mathbb{E}_{X, X' \sim p} \left[ k(X, X') \right], \tag{6}$$

where $k(X, X')$ is the Gaussian or Radial Basis Function (RBF) kernel (Borodachov et al., 2019). Wang & Isola (2020) showed that this objective is optimized by uniform distribution. Similarly Trosten et al. (2023) designed $\mathcal{L}_{\text{uniform}}$ and interpret it as a negative entropy on the hypersphere.

## 3 OPTIMIZING FOR DISPERSION

All objectives discussed in §2 are pairwise-based objectives, meaning that they require calculation of the full pairwise distance matrix, which scales poorly with the growth of $N$ and $d$. Moreover, Max-Min and KoLeo consider only the point and it's nearest neighbor for each update. We give a new interpretation of the Uniformity regularizer discussed in §2, in terms of (squared) MMD. Second, we define Lloyd and Sliced objectives that approximate optimal dispersion without requiring the full pairwise distance matrix. It makes those two objectives more suitable for large-scale parameter optimization.

### 3.1 PAIRWISE REGULARIZERS AND MMD

The distribution of perfectly dispersed embeddings is similar to a uniform distribution on the hypersphere. Dispersing embeddings can then be seen as minimizing the 'distance' between the embedding distribution and the uniform distribution $\text{Unif}(\mathbb{S}_d)$. The Raleigh test for uniformity is not well suited for this purpose as discussed in the previous section. An alternative statistical test for uniformity can be derived from the *maximum mean discrepancy* (MMD), which measures the distance between two probability distributions (Gretton et al., 2012). Lemma 1 implies that the squared MMD between the distribution of the embeddings and the uniform distribution on the sphere can be computed using embeddings only, up to a constant.

> **Lemma 1** ($\text{MMD}^2$ **and spherical embeddings.**) *Let $p$ be any distribution on $\mathbb{S}_d$ and let $k$ be a kernel on $\mathbb{S}_d$ such that $k(x, y) = f(\langle x, y \rangle)$ for some function $f \colon [-1, 1] \to \mathbb{R}$. Assume all random variables are independent.*
>
> *Up to a normalizing constant $c \in \mathbb{R}$, we have*
>
> $$\text{MMD}^2[p, \text{Unif}(\mathbb{S}_d)] = \mathbb{E}_{X, X' \sim p} \left[ k(X, X') \right] - c.$$

The proof of Lemma 1 is deferred to Appendix A.1. Using the radial basis function kernel $k(x, y) = \exp\left(-\lambda \|x - y\|^2\right)$ in the result of Lemma 1, we see that minimizing the estimated squared MMD of the embeddings and the uniform distribution is equivalent to minimizing

$$\mathcal{L}_{\text{MMD}} = \frac{1}{n(n-1)} \sum_{i=1}^{n} \sum_{\substack{j=1 \\ i \neq j}}^{n} \exp\left(\gamma \langle x_i, x_j \rangle\right), \tag{7}$$

where $X \subseteq \mathbb{S}_d$ is a set of $n$ embeddings and $\gamma := 2\lambda > 0$. The intuition for $\mathcal{L}_{\text{MMD}}(X)$ is that the embeddings are pushed away from each other when minimizing $\mathcal{L}_{\text{MMD}}(X)$, thereby improving the uniformity of the embedding distribution. The parameter $\gamma$ determines the emphasis on the distance between embeddings, *i.e.*, a larger $\gamma$ results in a larger emphasis on close embeddings.

The regularizer $\mathcal{L}_{\text{MMD}}$ is related to the partial loss function used by Trosten et al. (2023) to disperse image representation embeddings for few shot learning, as well as the energy-based approaches to Tammes and Thompson problem (Gautam & Vaintrob, 2013; Liu et al., 2018b; 2021). In particular, the exponential of the energy optimized by Trosten et al. (2023); Wang & Isola (2020) differs from $\mathcal{L}_{\text{MMD}}$ by a constant. Our work thus provides a new justification of their objective.

## 3.2   LLOYD'S ALGORITHM

An alternative formulation of dispersion comes from casting maximal dispersion as *quantization* of a uniform measure. Quantization refers to the problem of approximating a given measure by an empirical measure supported at a few centers. When the given measure is uniform over some support set, the optimal centers are spread out uniformly over the support; and can be calculated by Lloyd's algorithm (Lloyd, 1982), henceforth *Lloyd*, which iteratively moves each centroid to the center of mass of its Voronoi cell. When the given measure is another empirical measure, quantization is equivalent to *k-means clustering*. When the space is Riemannian and not Euclidean, both quantization and clustering generalize readily with an adequate choice of distance (Le Brigant & Puechmorel, 2019). While Lloyd's algorithm and *k*-means are originally batch algorithms, stochastic gradient versions have been developed (Bottou & Bengio, 1995; Sculley, 2010), including, independently, in the Riemannian case (Le Brigant & Puechmorel, 2019). In general, given a domain $\mathbb{D}$, which could be a manifold or a compact subset of one (for quantization), or a discrete dataset (for clustering), the $n$ optimal centroids are a minimizer of[2]

$$\mathcal{L}_{\text{Lloyd}} = \mathbb{E}_{\mathsf{Y} \sim \text{Unif}(\mathbb{D})} \left[ \min_{j \in [n]} \frac{1}{2} d^2(\mathsf{Y}, x_j) \right]. \tag{8}$$

A stochastic gradient of the Lloyd regularizer can be obtained by drawing $m$ uniform samples on $\mathbb{D}$. Intuitively, each cluster center is pulled toward the barycenter of the uniform samples assigned to it; an approximation to the true Voronoi barycenter.

For dispersion on the sphere, we take $\mathbb{D} = \mathbb{S}_d$. While traditionally Lloyd's algorithm corresponds to minimizing $\mathcal{L}_{\text{Lloyd}}$ alone, we propose using $\mathcal{L}_{\text{Lloyd}}$ as a regularizer to move $X$ closer to optimal Voronoi centers of the sphere, while also minimizing some main task-specific objective. The complexity of this regularizer is controlled by the number of samples: For efficiency, $m$ should be much less than $n$, in which case most cluster centers are not updated in an iteration. However, unlike for MMD, the stochastic gradient takes into account all of $X$ through the cluster assignment.

## 3.3   SLICED DISPERSION

The previously discussed algorithms are generally applicable to other manifolds. We now show how using properties of the sphere we may obtain an alternative algorithm for embeddings dispersion. The key idea is that, while in 2 or more dimensions it is hard to find the location of $n$ evenly distributed points, on $\mathbb{S}_1$ this can be done efficiently: The following set of angles is one optimal configuration:

$$\Phi = (\phi_1, \dots, \phi_n) \quad \text{where} \quad \phi_k = -\pi \frac{n+1}{n} + \frac{2\pi k}{n}.$$

Any other optimal configuration must be a rotation of this one, *i.e.* $\tau + \Phi$ for $\tau \in (-\pi, \pi)$. followed by a permutation of these angles. Given a permutation $\sigma \in \Pi_n$ denote $\Phi_\sigma = (\phi_{\sigma(1)}, \dots, \phi_{\sigma(n)})$. We can then write the set of all possible ordered optimally-dispersed configurations as

$$D_n \mathbb{S}_1 := \{\tau + \Phi_\sigma \mid \tau \in (-\pi, \pi), \sigma \in \Pi_n\}. \tag{9}$$

Given an ordered configuration of angles $\Theta = (\theta_1, \dots, \theta_n) \subset \mathbb{S}_1$, we define its (angular) distance to the maximally-dispersed set as:

$$d^2(\Theta, D_n \mathbb{S}_1) = \min_{\hat{\Theta} \in D_n \mathbb{S}_1} \sum_{i=1}^n \frac{1}{2}(\theta_i - \hat{\theta}_i)^2. \tag{10}$$

---

[2]More generally, the target measure need not be uniform. Le Brigant & Puechmorel (2019) discuss more general conditions for the existence of a minimizer.

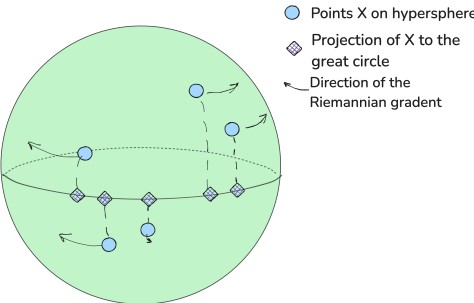

**Figure 1:** Visualization of a single update in sliced dispersion, for a great circle $\mathbb{S}_{pq}$. Sliced dispersion maximizes dispersion in expectation over all great circles.

Lemma 3 defined and proved in Appendix A.2 shows that any configuration of angles can be efficiently projected to its nearest maximally-dispersed configuration. We defer all proofs in this section to Appendix A.2.

In arbitrary dimensions, a similar construction is not possible, since the optimal configurations do not have tractable characterizations. We instead *slice* a high-dimensional spherical dataset along a great circle; similar to Bonet et al. (2023). The following result gives the geodesic projection.

> **Lemma 2 (Projection onto great circle.)** Let $p, q \in \mathbb{S}_d$ with $\langle p, q \rangle = 0$. *Two such vectors determine a unique great circle $\mathbb{S}_{pq} \subset \mathbb{S}_d$ defined by:*
>
> $$\mathbb{S}_{pq} := \{\cos(\theta)p + \sin(\theta)q \mid -\pi \le \theta < \pi\} \simeq \mathbb{S}_1.$$
>
> *The nearest point on $\mathbb{S}_{pq}$ to a given $x \in \mathbb{S}_d$ is:*
>
> $$\text{proj}_{\mathbb{S}_{pq}}(x) = \arctan2\left(\langle x, q \rangle, \langle x, p \rangle\right). \tag{11}$$

A well-dispersed configuration over $\mathbb{S}_d$ should remain fairly well-dispersed along any slice on average. If we denote $\text{proj}_{\mathbb{S}_{pq}}(X) := (\text{proj}_{\mathbb{S}_{pq}}(x_1), \ldots \text{proj}_{\mathbb{S}_{pq}}(x_n))$, we may capture this intention by the following **sliced dispersion regularizer**:

$$\mathcal{L}_{\text{Sliced}} = \mathbb{E}_{p,q}\left[d^2(\text{proj}_{\mathbb{S}_{pq}}(X), D_n\mathbb{S}_{pq})\right], \tag{12}$$

where $d^2$ is defined in eq. (10), and the expectation is over orthogonal pairs $p, q$. Note that unlike algorithms such as principal geodesic analysis (Fletcher et al., 2004), which keep $X$ fixed but optimize for some $p, q$ to maximize variance, our intuition is the opposite: we want to update $X$ in order to increase dispersion along *any* great circle. The following proposition efficiently computes stochastic gradients of $\mathcal{L}_{\text{Sliced}}$.

> **Proposition 1** *Denote $\theta_i^{pq} = \text{proj}_{\mathbb{S}_{pq}}(x_i)$, and $\hat{\theta}_i^{\star pq}$ the corresponding dispersion maximizer computed using Lemma 3. The Riemannian gradient of $\mathcal{L}_{Sliced}$ is given by:*
>
> $$\text{grad}_{x_i}\mathcal{L}_{Sliced} = \mathbb{E}_{p,q}\left[(\theta_i^{pq} - \hat{\theta}_i^{\star pq})\frac{\langle x_i, p \rangle q - \langle x_i, q \rangle p}{\langle x_i, q \rangle^2 + \langle x_i, p \rangle^2}\right].$$

## 3.4 RIEMANNIAN OPTIMIZATION ON HYPERSPHERE

Optimization for dispersion can be defined as constrained optimization problem in $\mathbb{R}$, where constraint is that points lie on the hypersphere. This can be solved by ignoring spherical constrains and projecting the parameters onto the sphere after the gradient update, however *convergence is not guaranteed*, because the sphere is not a convex set, even though it can give acceptable results with careful initialization (Raman & Yang, 2019). Alternatively, we can rely on Riemannian optimization (Bonnabel, 2013; Becigneul & Ganea, 2019) in $\mathbb{S}^{d-1}$ as effective *unconstrained* extension (Bloch, 2015; Boumal, 2023) with guaranteed convergence (Bonnabel, 2013). We further empirically explore the convergence of both methods in Appendix B.

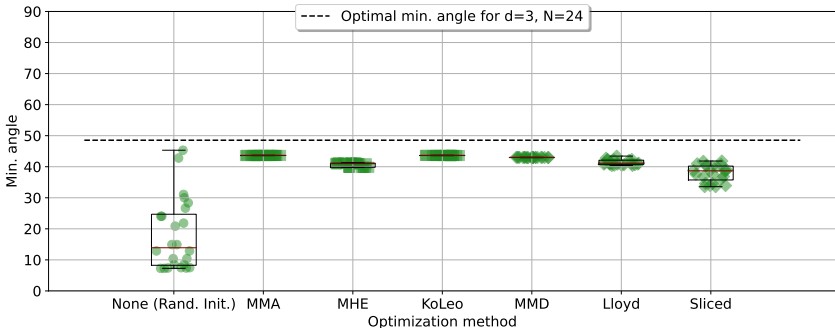

**Figure 2:** Minimum angles (degrees) for each of the N=24 points with respect to optimization methods. `Optimal Solution` shows the angle for known optimal solution. `Rand.Init.` represents the points generated uniformly at random on the surface of the sphere. All optimizations start with the `Rand.Init.` as an initialization. Optimal minimum angle is equal to $48.53529763°$. An ideal configuration is achieved when all angles are equal to optimal angle.

## 4 APPLICATIONS

We demonstrate the application of dispersion objectives and provide a comparative analysis on both synthetic and real-world tasks. Unlike previous studies, we employ Riemannian optimization (Bonnabel, 2013; Becigneul & Ganea, 2019) directly on the hypersphere using geoopt[3] (Kochurov et al., 2020), instead of relying on projection onto the hypersphere at each gradient step as discussed in §3.4.

### 4.1 TAMMES PROBLEM

We evaluate the dispersion methods introduced in §2 and §3 by verifying that they can approximate the known solution to the Tammes problem for $N = 24$ in three dimensions (Robinson, 1961), by considering the minimum angle between points of the optimal configuration. Uniformly sampled points are dispersed using the regularizers described in §2 and §3. Optimization is done with Riemannian Adam for 2.5k epochs. The MMD regularizer was minimized with $\gamma = 25$. The sliced dispersion regularizer used a single randomly generated pair of axes during each epoch. The Lloyd regularizer was used with 300 samples. We set $s = 0$ for MHE. All regularizers were used with learning rate $5 \cdot 10^{-3}$.

The minimum angles of the points distributed using the MMD, MMA and KoLeo regularizers are close to the optimal minimum angle for all presented $N$ as shown in Figure 2. The Lloyd and MHE regularizers follows closely, but seems to approximate the solutions less accurately. The sliced dispersion regularizer, however, seems to approximate the solutions worse than the other regularizers. More results on Tammes problem approximation can be found in Appendix C.

### 4.2 SYNTHETIC EMBEDDINGS

In practice, we are mostly interested in dispersion of large amount of points in dimension $d \gg 3$. Text embeddings can be a particular example of the set of points that can benefit from dispersion (Tokarchuk & Niculae, 2024). One can argue that dispersion connects strongly to the dimensionality, and in higher dimension embeddings are dispersed naturally. However, higher dimensionality comes with higher computation and memory cost. Also, there is no guarantee that space is occupied efficiently. Thus, Gao et al. (2019) showed that representation in vanilla Transformer Vaswani et al. (2017) occupies only part of the whole space. We evaluate the behaviour of the regularizers discussed in §2 with synthetic embedding by generating matrix containing 20k embeddings in $d = 128$. The data was generated by sampling a matrix entry-wise from a `PowerSpherical`(De Cao & Aziz, 2020) distribution with $\kappa$ equal to 100. This exemplifies a scenario where the embeddings are well spread out from the beginning. The regularizers were minimized using Riemannian Adam (Kingma & Ba,

---
[3]https://github.com/geoopt/geoopt

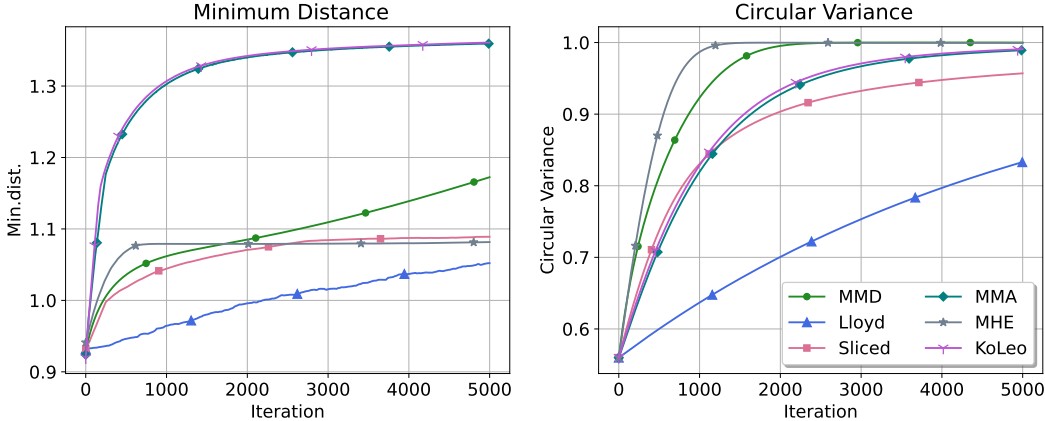

**Figure 3:** Comparison of different dispersion objectives on synthetic data.

2015; Becigneul & Ganea, 2019; Kochurov et al., 2020), for 5k iterations with learning rate $1 \cdot 10^{-3}$. We set $\gamma$ of the MMD regularizer to 10.0, number of samples for Lloyd to 8192. Due to the hardware constraints we implement batched version of MHE and MMA, and use batch size equal to 16K. We set $s = 0$ for MHE. We also rely on the batched version of axis-aligned Sliced regularizer with batch size equal to 128.

Figure 3 shows the minimum distance and circular variance for various regularizers. KoLeo and MMA performs the best in terms of minimum distance, with MMD being second best. MMD and MHE reach the highest circular variance, followed closely by MMA and KoLEo. It it important to note, that reaching the best minimum distance and/or circular variance does not necessarily mean the best performance on the downstream task. The trade-off between performance and dispersion should be considered for each particular case.

### 4.3 IMAGE CLASSIFICATION WITH PROTOTYPES

| prototypes | 50 | | 100 | | 200 | |
|---|---|---|---|---|---|---|
| | Acc. | $d_{\min}$ | Acc. | $d_{\min}$ | Acc. | $d_{\min}$ |
| MMCS (+projection) | 41.67 | 1.22 | 42.76 | 1.36 | 43.03 | 1.44 |
| MMCS ($\mathbb{S}$) | 42.59 | 1.46 | 42.96 | 1.52 | **43.27** | 1.56 |
| MMA ($\mathbb{S}$) | 41.72 | 1.39 | **43.47** | 1.46 | 42.90 | 1.51 |
| MHE ($\mathbb{S}$) | 43.37 | 1.41 | 42.25 | 1.6 | 34.47 | 1.58 |
| KoLeo ($\mathbb{S}$) | 41.78 | 1.37 | 43.12 | 1.44 | 42.37 | 1.49 |
| MMD ($\gamma = 1, \mathbb{S}$) | **43.87** | 1.22 | 42.73 | 1.57 | 34.53 | 1.58 |
| Lloyd (samples=200, $\mathbb{S}$) | 41.69 | 1.20 | 42.42 | 1.30 | 43.09 | 1.35 |
| Sliced ($\mathbb{S}$) | 40.76 | 1.10 | 42.34 | 1.20 | 42.92 | 1.33 |

**Table 1:** ImageNet-200 classification accuracy. Prototypes are trained with different separation conditions. MMCS refers to the setup of Mettes et al. (2019). In bold we emphasise the best accuracy in a column.

Mettes et al. (2019) showed that learning prototypes with dispersion encouraged by minimizing the maximum cosine similarity (MMCS) on hypersphere improves classification results on ImageNet-200. We first show in Table 1 that applying Riemannian optimization rather than re-normalizing parameters after each gradient update as in Mettes et al. (2019) leads to the better class separation, and as a result better classification accuracy. Second, we compare the classification accuracy given the prototypes trained with different dispersion objectives discussed in §2 and §3. We use unconstrained optimization on the sphere for all methods, and results with projection is shown only for comparison. Also, Table 1 shows that when prototypes dimension is equal 50, MMD performs the best among all dispersion objectives, even though the minimum distance is smaller compared to other pairwise-

distance based objectives. It proves that even though we can measure the dispersion using minimum distance, we cannot rely on this metric alone as a predictive factor of the downstream task accuracy.

Interestingly, when dimensionality is equal to the number of points, MMD and MHE prototypes results degrade significantly. For both MMD and MHE minimum distance and median distance are equal to exactly 1.5758213996887207 radian or $90.3°$, which resembles orthogonal solution. Since the network is trained with the squared cosine distance, when angle between two points is $90°$, the distance is equal to exactly 1 to all possible prototypes, which makes the loss less informative. Results reported by Mettes et al. (2019) also confirms that one-hot embeddings (orthogonal solution) perform badly on the task at hand.

### 4.4 Neural Machine Translation

Embeddings learned with the vanilla transformer model (Vaswani et al., 2017) are known for their inefficiency in utilizing space effectively, leading to the collapse of token representations (Gao et al., 2019; Wang et al., 2020a). This issue is particularly pronounced for rare tokens (Gong et al., 2018; Tokarchuk & Niculae, 2024; Zhang et al., 2022). Gong et al. (2018) proposed to alleviate the problem of rare tokens by learning frequency-agnostic embeddings, while Zhang et al. (2022) proposed to use contrastive learning. In our approach, we tackle this challenge by focusing on the concept of dispersion. Specifically, we train a Neural Machine Translation (NMT) system and jointly optimize the decoder embeddings to enhance their dispersion.

$$\mathcal{L}(\mathbf{W}, \mathbf{E}_Y) = \mathcal{L}_{\text{MT}}(\mathbf{W}, \mathbf{E}_Y) + \lambda \mathcal{L}_{\text{disp}}(\mathbf{E}_Y) \tag{13}$$

We report results on two WMT translation tasks[4]: WMT 2016 Romanian→English (ro-en) with 612K training samples and WMT 2019 English→German (en-de) with 9.1M training samples (including back-translated data). We measure translation accuracy on the best checkpoint according to validation BLEU score using SacreBLEU (Papineni et al., 2002; Post, 2018) and COMET (Rei et al., 2020). Detailed training parameters are discussed in Appendix D.

Table 2 shows the BLEU and COMET results on newstest2016 for ro-en and newstest2016 en-de along with the dispersion metrics. Similarly to image classification, doing Riemannian optimization in order to disperse embeddings leads to better dispersion and higher BLEU and COMET scores.

| model | ro-en | | | | en-de | | | |
|---|---|---|---|---|---|---|---|---|
| | BLEU | COMET | $d_{\min}$ | svar | BLEU | COMET | $d_{\min}$ | svar |
| euclidean baseline | 31.4 | 0.790 | 0.003 | 0.19 | 33.1 | 0.819 | 0 | 0 |
| spherical baseline | 32.2 | 0.793 | 0.001 | 0.57 | 33.7 | **0.825** | 0.001 | 0.408 |
| +MMD | 32.3 | **0.795** | 0.001 | 0.56 | **33.9** | **0.825** | 0.001 | 0.410 |
| +Lloyd | **32.4** | 0.791 | 0.001 | 0.60 | 33.4 | 0.822 | 0.001 | 0.414 |
| +Sliced | **32.4** | **0.795** | 0.435 | 0.99 | 33.5 | 0.820 | 0.222 | 0.999 |

**Table 2:** newstest2016 ro-en and en-de results on discrete NMT. Embeddings are 128 dim.

We investigate the effect of Riemannian optimization by analyzing the gradient norm of the Euclidean baseline (vanilla transformer) and the Spherical baseline, as shown in Figure 4a, alongside the minimum pairwise distance for each embedding, presented in Figure 4b. The results reveal that the gradient norm for the Riemannian approach is approximately ten times higher than that of the Euclidean baseline. We hypothesize that this increased gradient norm contributes to better dispersion of rare tokens, thereby mitigating representation collapse. They dynamics of gradient norms and minimum distances can be seen in Appendix E.

## 5 Continuous-Output Neural Machine Translation

Continuous-Output NMT (CoNMT, Kumar & Tsvetkov, 2019) reformulates machine translation as a sequential continuous regression problem of predicting the embedding of the next word, instead of the more usual discrete classification formulation. Tokarchuk & Niculae (2024) recently showed that

---

[4]https://www2.statmt.org/

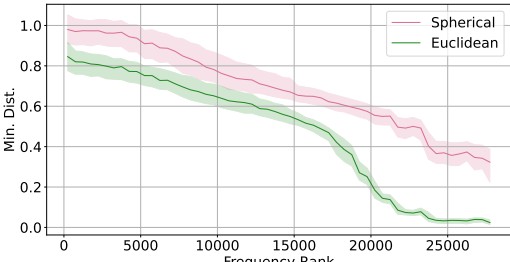

**(a)** Gradient norms of the embeddings for trained (step equal 40000) Spherical and Euclidean NMT baselines. Frequency rank refers to the position of the token in the vocubulary, where most frequent token has rank 0 and lest frequent rank vocabulary size.

**(b)** Minimum distance to the nearest embeddings for trained (step equal 40000) Spherical and Euclidean baselines. Frequency rank refers to the position of the token in the vocubulary, where most frequent token has rank 0 and lest frequent rank vocabulary size.

**Figure 4:** Embeddings matrix gradient norms (a) and minumum distances (b) for Euclidean and Spherical baselines.

dispersion plays an important role and greatly impacts performance. We follow closely their setup and apply the dispersion regularizers in order to achieve dispersion. Pre-trained embeddings come from the well-trained discrete model. We present results for WMT 2016 `ro-en` with 612k training samples. Table 3 shows the BLEU score results on `newstest2016` for CoNMT models with different target embeddings $\mathbf{E_Y}$, alongside dispersion measures defined in §2.2

We conduct two types of experiments. First we train a vanilla transformer model (Vaswani et al., 2017). Resulting embeddings are in Euclidean space, so we project it onto the sphere by dividing to the norms of embeddings. To spread out the embeddings we then use Riemannian optimization on the sphere with `geoopt` (Kochurov et al., 2020) using three different regularizers. We refer to this as 'offline' methods in Table 3. Second, we train transformer model with embeddings explicitly modeled to be on the sphere using Riemannian optimization. In this case, we can apply dispersion regularizers directly during optimization. Discrete models that were used to extract embeddings are the same as in Table 2.

Spreading out the projected embeddings results into the BLEU score improvement with MMD and Sliced dispersion. For all dispersion regularizers, we can see that $\text{svar}(\mathbf{E_Y})$ is increasing. However, $d_{\min}(\mathbf{E_Y})$ decreases for the Lloyd regularizer, which seemingly also impacts the BLEU score.

When adding dispersion regularizers, there are no significant fluctuations in $\text{svar}(\mathbf{E_Y})$, except for the Sliced regularizer. We leave thorough investigation of the observed behaviour for the future work.

| Tgt. Emb. $\mathbf{E_Y}$ | $\text{svar}(\mathbf{E_Y}) \uparrow$ | $d_{\min}(\mathbf{E_Y}) \uparrow$ | BLEU$\uparrow$ |
|---|---|---|---|
| euclidean (proj.) | 0.191 | 0.014 | 27.8 |
| +offline MMD | 0.599 | 0.372 | 29.7 |
| +offline Lloyd | 0.585 | 0.004 | 27.7 |
| +offline Sliced | 0.979 | 0.106 | 29.6 |
| spherical | 0.57 | 0.001 | 29.9 |
| +MMD | 0.56 | 0.001 | 30.0 |
| +Lloyd | 0.60 | 0.001 | **30.1** |
| +Sliced | **0.99** | **0.435** | 30.0 |

**Table 3:** Impact of the dispersion of the target embeddings on the CoNMT results. We report BLEU scores on the `newstest2016` for `ro-en`. Beam size is equal to 5.

## 6 CONCLUSION

In this work, evaluate several dispersion objectives on the hypersphere, including one that is equivalent to the widely used Maximum Mean Discrepancy (MMD) method, as well as two novel approaches: Lloyd and Sliced. We compare these objectives against various pairwise distance-based methods previously explored in the literature. Our experimental results show that these methods can approximate the Tammes problem solution, and also allow improvement on few-shot Image Classification with prototypes, machine translation and the CoNMT task, which uses cosine distance both for training and decoding.

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

# A   APPENDIX

## A.1   MMD DISPERSION: PROOFS

### A.1.1   MMD$^2$ AND SPHERICAL EMBEDDINGS: PROOF OF LEMMA 1

The squared MMD of two probability distributions $p$ and $q$ is equal to (Gretton et al., 2012, Lemma 6)

$$\mathrm{MMD}^2[p, q] = \mathbb{E}_{\mathsf{X},\mathsf{X}'\sim p}[k(\mathsf{X}, \mathsf{X}')] - 2\mathbb{E}_{\mathsf{X}\sim p,\mathsf{Y}\sim q}[k(\mathsf{X}, \mathsf{Y})] + \mathbb{E}_{\mathsf{Y},\mathsf{Y}'\sim q}[k(\mathsf{Y}, \mathsf{Y}')].$$

We show that the last two expectations are constant, when $p$ is a distribution on the hypersphere $\mathbb{S}_d$ and $q$ is $\mathrm{Unif}(\mathbb{S}_d)$. Let $z, z' \in \mathbb{S}_d$ and let $Q$ be a rotation matrix such that $Qz = z'$. Note that $\mathsf{Y} \sim \mathrm{Unif}(\mathbb{S}_d)$ if and only if $Q^\top \mathsf{Y} \sim \mathrm{Unif}(\mathbb{S}_d)$, and $\langle Qz, z \rangle = \langle z, Q^\top z \rangle$. It then follows that

$$\mathbb{E}_{\mathsf{Y}\sim \mathrm{Unif}(\mathbb{S}_d)}[k(z, \mathsf{Y})] = \mathbb{E}_{\mathsf{Y}\sim \mathrm{Unif}(\mathbb{S}_d)}[k(z', \mathsf{Y})],$$

since $k(x, y) = f(\langle x, y \rangle)$. Hence, there exists a $c \in \mathbb{R}$ such that for all $z \in \mathbb{S}_d$ we have

$$\mathbb{E}_{\mathsf{Y}\sim \mathrm{Unif}(\mathbb{S}_d)}[k(z, \mathsf{Y})] = c.$$

Consequently, $\mathbb{E}_{\mathsf{X}\sim p,\mathsf{Y}\sim \mathrm{Unif}(\mathbb{S}_d)}[k(\mathsf{X}, \mathsf{Y})] = c$ and $\mathbb{E}_{\mathsf{Y},\mathsf{Y}'\sim \mathrm{Unif}(\mathbb{S}_d)}[k(\mathsf{Y}, \mathsf{Y}')] = c$. The desired result follows immediately.

## A.2   SLICED DISPERSION: PROOFS

### A.2.1   OPTIMAL 1-D DISPERSION

**Lemma 3** *Optimal 1-d dispersion.* The projection

$$\arg\min_{\hat{\Theta}\in D_n\mathbb{S}_1} \sum_{i=1}^n \frac{1}{2}(\theta_i - \hat{\theta}_i)^2$$

is given by $\hat{\theta}_i^\star = \tau^\star + \phi_{\sigma^{-1}(i)}$, where $\sigma$ is the permutation s.t. $\theta_{\sigma(1)} \leq \theta_{\sigma(2)} \leq \ldots \leq \theta_{\sigma(n)}$, and $\tau^\star = \frac{\sum_i \theta_i}{n}$. The projection can be calculated in $O(n \log n)$, the dominating cost being sorting the angles.

We aim to prove the assertion that the projection

$$\arg\min_{\hat{\Theta}\in D_n\mathbb{S}_1} \sum_{i=1}^n \frac{1}{2}(\theta_i - \hat{\theta}_i)^2$$

is given by $\hat{\theta}_i^\star = \tau^\star + \phi_{\sigma^{-1}(i)}$, where $\sigma$ is the permutation st $\theta_{\sigma(1)} \leq \theta_{\sigma(2)} \leq \ldots \leq \theta_{\sigma(n)}$, and $\tau^\star = \frac{\sum_i \theta_i}{n}$.

By definition, per eq. (9), $\hat{\Theta} = \tau + \Phi_\sigma$ and thus we may write the problem equivalently as

$$\arg\min_{\tau\in[-\pi,\pi),\sigma\in\Pi_n} \sum_i \frac{1}{2}\left(\theta_i - \phi_{\sigma(i)} - \tau\right)^2.$$

**Finding the permutation.** In terms of $\sigma$ the objective takes the form $-\sum_i \theta_i \phi_{\sigma(i)} + \text{const}$, so we must find the permutation that maximizes $\sum_i \theta_i \phi_{\sigma(i)} = \sum_i \theta_{\sigma^{-1}(i)} \phi_i$. By the rearrangement inequality (Hardy et al., 1952, Thms. 368–369), since $\phi_i$ is in ascending order, this sum is maximized when $\theta_{\sigma^{-1}(i)}$ is in ascending order; so the optimal $\sigma$ must be the inverse of the permutation that sorts $\Theta$.

**Finding $\tau$.** Ignore the constraints momentarily, and set the gradient of the objective to zero:

$$\frac{\partial}{\partial \tau} \sum_i \frac{1}{2}(\theta_i - \phi_{\sigma(i)} - \tau)^2 = \sum_i (\tau + \phi_{\sigma(i)} - \theta_i) = 0, \quad \text{implying} \quad n\tau = \sum_i \theta_i - \sum_i \phi_i = \sum_i \theta_i,$$

the last equality by choice of the zero-centered reference configuration $\Phi$. Since all $\theta_i \in [-\pi, \pi)$, so is their average, and thus the constraints are satisfied, concluding the proof.

### A.2.2 PROJECTION ONTO A GREAT CIRCLE

The projection we seek to compute is

$$\text{proj}_{\mathbb{S}_{pq}}(x) := \arg\min_{-\pi \leq \theta < \pi} d^2((\cos(\theta)p + \sin(\theta)q, x).$$

Since the geodesic distance satisfies $d^2(\cdot, \cdot) = \arccos\langle\cdot, \cdot\rangle$ and $\arccos$ is strictly decreasing on $(-1, 1)$, we have

$$\text{proj}_{\mathbb{S}_{pq}}(x) := \arg\max_{-\pi \leq \theta < \pi} \langle\cos(\theta)p + \sin(\theta)q, x\rangle.$$

As a side note, this shows that it doesn't matter whether we use geodesic or Euclidean distance to define this projection. Setting the gradient to zero yields

$$\cos(\theta)\langle q, x\rangle = \sin(\theta)\langle p, x\rangle,$$

or equivalently $\tan(\theta) = \langle q, x\rangle / \langle p, x\rangle$. The unique solution on $[-\pi, \pi)$ is given by the two-argument arctangent function (arctan2), also known as the argument of complex number $\langle p, x\rangle + i\langle q, x\rangle$ (Wikipedia contributors, 2024).

### A.2.3 GRADIENT OF SLICED DISTANCE

We first compute the Euclidean gradient of the desired expression:

$$\nabla_{x_i} \mathcal{L}_{\text{Sliced}}(X) = \nabla_{x_i} \mathbb{E}_{p,q}\left[d^2(\text{proj}_{\mathbb{S}_{pq}}(X), D_n\mathbb{S}_{pq})\right]. \tag{14}$$

First, by writing

$$d^2(\Theta, D_n\mathbb{S}_{pq}) = \min_{\hat{\Theta}} \sum_i \frac{1}{2}(\theta_i - \hat{\theta}_i)^2$$

we see this may be interpreted as an Euclidean projection and

$$\frac{\partial}{\partial \theta_i} d^2(\Theta, D_n\mathbb{S}_{pq}) = (\theta_i - \theta_i^\star).$$

But $\theta_i = \text{proj}_{\mathbb{S}_{pq}}(x_i)$ and we can write

$$\frac{\partial \theta_i}{\partial x_i} = \frac{\partial}{\partial x_i} \text{proj}_{\mathbb{S}_{p,q}}(x_i)$$

$$= \frac{\partial \theta_i}{\partial x_i} \tan^{-1}\left(\frac{\langle q, x\rangle}{\langle p, x\rangle}\right)$$

$$= \frac{\langle p, x\rangle q - \langle q, x\rangle p}{\langle q, x\rangle^2 + \langle p, x\rangle^2}.$$

Putting the two together via the chain rule yields

$$\nabla_{x_i} \mathcal{L}_{\text{Sliced}}(X) = (\theta_i^{pq} - \hat{\theta}_i^{\star pq}) \frac{\langle p, x_i\rangle q - \langle q, x_i\rangle p}{\langle q, x_i\rangle^2 + \langle p, x_i\rangle^2}. \tag{15}$$

Notice that the second term is a vector in $\mathbb{R}^{d+1}$ that is orthogonal to $x_i$ because:

$$\langle x_i, \langle p, x_i\rangle q - \langle q, x_i\rangle p\rangle = \langle p, x_i\rangle\langle q, x_i\rangle - \langle q, x_i\rangle\langle p, x_i\rangle = 0.$$

Therefore,

$$\text{grad}_{x_i} \mathcal{L}_{\text{Sliced}}(X) = \nabla_{x_i} \mathcal{L}_{\text{Sliced}}(X).$$

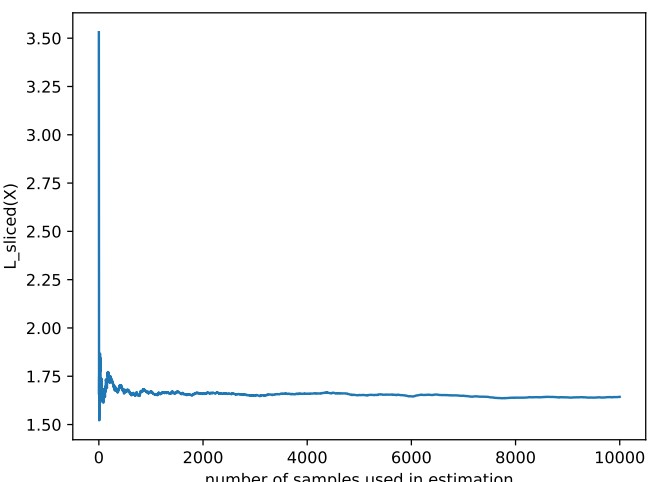

**Figure 5:** Convergence of the sliced regularizer.

### A.3 CONVERGENCE OF THE SLICED REGULARIZER

Figure Figure 5 shows that with the approximately 1000 samples Sliced regularizer reaches convergence.

## B RIEMANNIAN VS EUCLIDEAN OPTIMIZATION

### B.1 TAMMES PROBLEM

We compare the results of optimal angle approximation using constrained optimization in $\mathbb{R}^d$ with projection Appendix B.1 and unconstrained Riemannian optimization in $\mathbb{S}^{d-1}$ Appendix B.1. We perform optimization with the same parameters in both cases which identical to parameters described in §4.1. We exclude Sliced from the comparison since in both cases custom Riemannian gradient is calculated. However, for all other methods except KoLeo we can clearly see that optimization in $\mathbb{R}^d$ fails to converge to the (sub)-optimal solution compared to unconstrained optimization in $\mathbb{S}^{d-1}$.

## C TAMMES PROBLEM: ADDITIONAL RESULTS

In we present additional approximation results for Tammes problem for $N = (13, 14, 128)$. For N=13 and N=14 we compare with the theoretically proven solutions (Musin & Tarasov, 2012; 2015), for N=128 we use numerical solution (Cohn, 2024).

## D NEURAL MACHINE TRANSLATION: EXPERIMENTAL SETUP

For subword tokenization we used the same SentencePiece (Kudo & Richardson, 2018) model, specifically the one used in the MBart multilingual model (Liu et al., 2020). This choice allows for unified preprocessing for all languages we cover. We used fairseq (Ott et al., 2019) framework for training our models. Baseline discrete models (eucledian baseline) are trained with cross-entropy loss, label smoothing equal to 0.1 and effective batch size 65.5K tokens. All models are trained with learning rate $5 \cdot 10^{-4}$ and 10k warm-up steps for 50k steps in total. Spherical baseline and models with dispersion regularizer are trained by defining decoder's embeddings layer as a manifold parameter and using Riemannian Adam (Becigneul & Ganea, 2019) with learning rate $5 \cdot 10^{-3}$. We used SacreBLEU (Post, 2018) with the following signa-

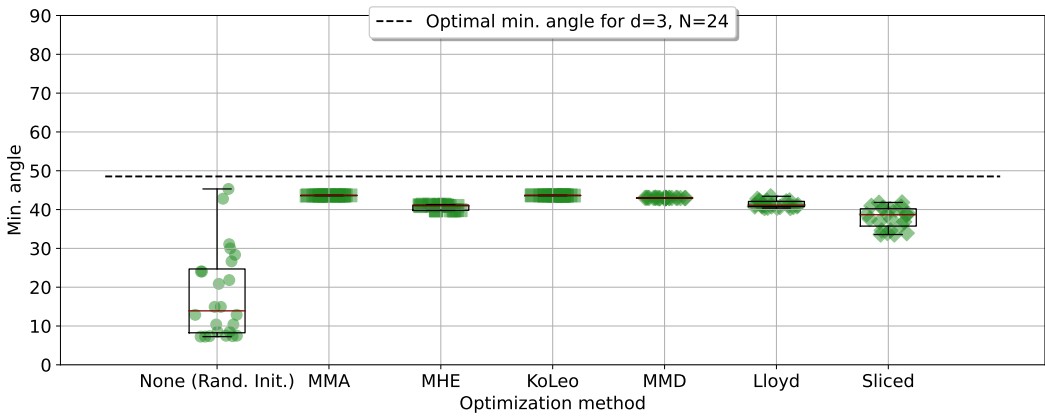

**(a)** Unconstrained optimization in $\mathbb{S}^{d-1}$.

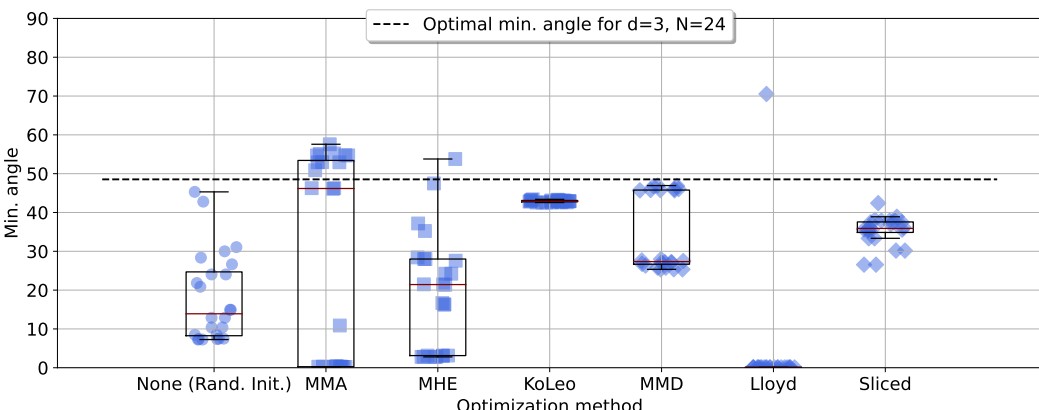

**(b)** Constrained optimization in $\mathbb{R}^d$ (projection).

**Figure 6:** Minimum angles (degrees) for each of the N=24 points with respect to optimization methods. `Optimal Solution` shows the angle for known optimal solution. `Rand.Init.` represents the points generated uniformly at random on the surface of the sphere. All optimizations start with the `Rand.Init.` as an initialization. Optimal minimum angle is equal to 48.53529763°. Ideal configuration is achieved when all angles equal to optimal angle, *i.e.*, lie on the optimal angle line. (a) refers to the Unconstrained optimization in $\mathbb{S}^{d-1}$, while (b) show results for Constrained optimization in $\mathbb{R}^d$ (projection).

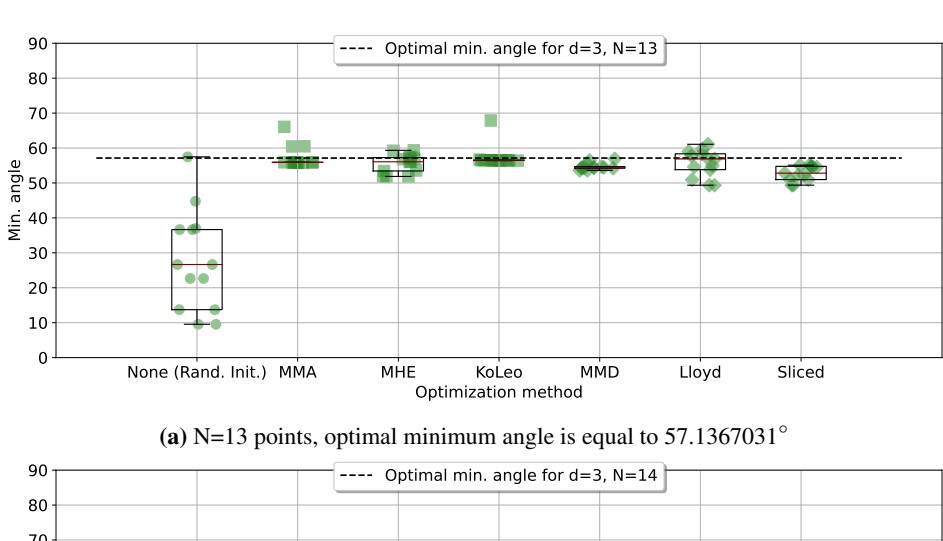

**(a)** N=13 points, optimal minimum angle is equal to 57.1367031°

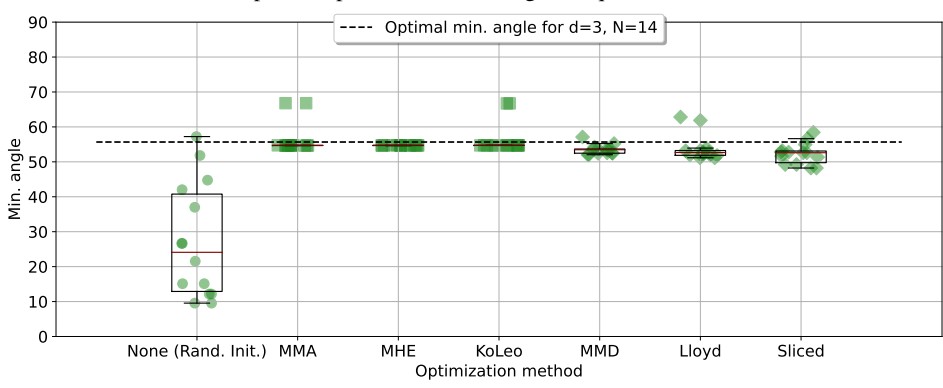

**(b)** N=14 points, optimal minimum angle is equal to 55.6705700°

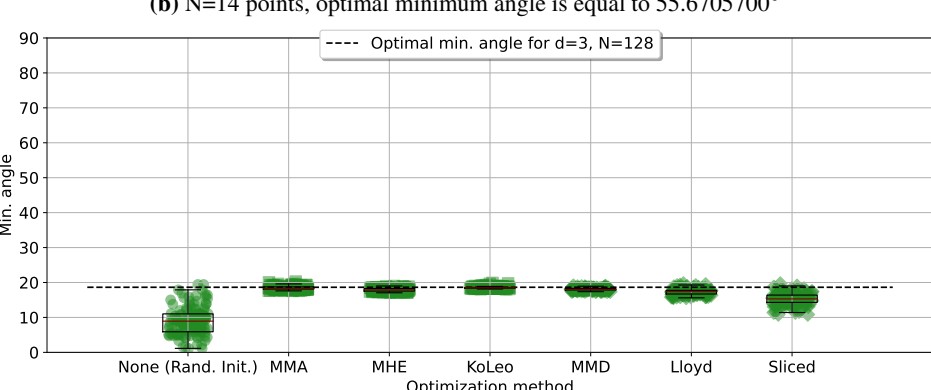

**(c)** N=128 points, optimal minimum angle is equal to 18.6349726°

**Figure 7:** Minimum angles (degrees) distributions for various points arrangements with d=3 and N=(13,14,128). `Optimal Solution` shows the angle for known optimal solution. `Rand.Init.` represents the points generated uniformly at random on the surface of the sphere. All optimizations start with the `Rand.Init.` as an initialization.

ture `nrefs:1|case:mixed|eff:no|tok:13a|smooth:exp|version:2.3.1` and COMET (Rei et al., 2020) with `unbabel-comet` library version 2.2.2[5] and `Unbabel-wmt22-comet-da` model.

---

[5] https://github.com/Unbabel/COMET

# E  NEURAL MACHINE TRANSLATION: GRADIENT NORMS

We show in Figure 8 how gradient norms and minimum distances of target language embeddings vary throughout the training process. Note that at the step=0, the norms and minimum distances are the same.

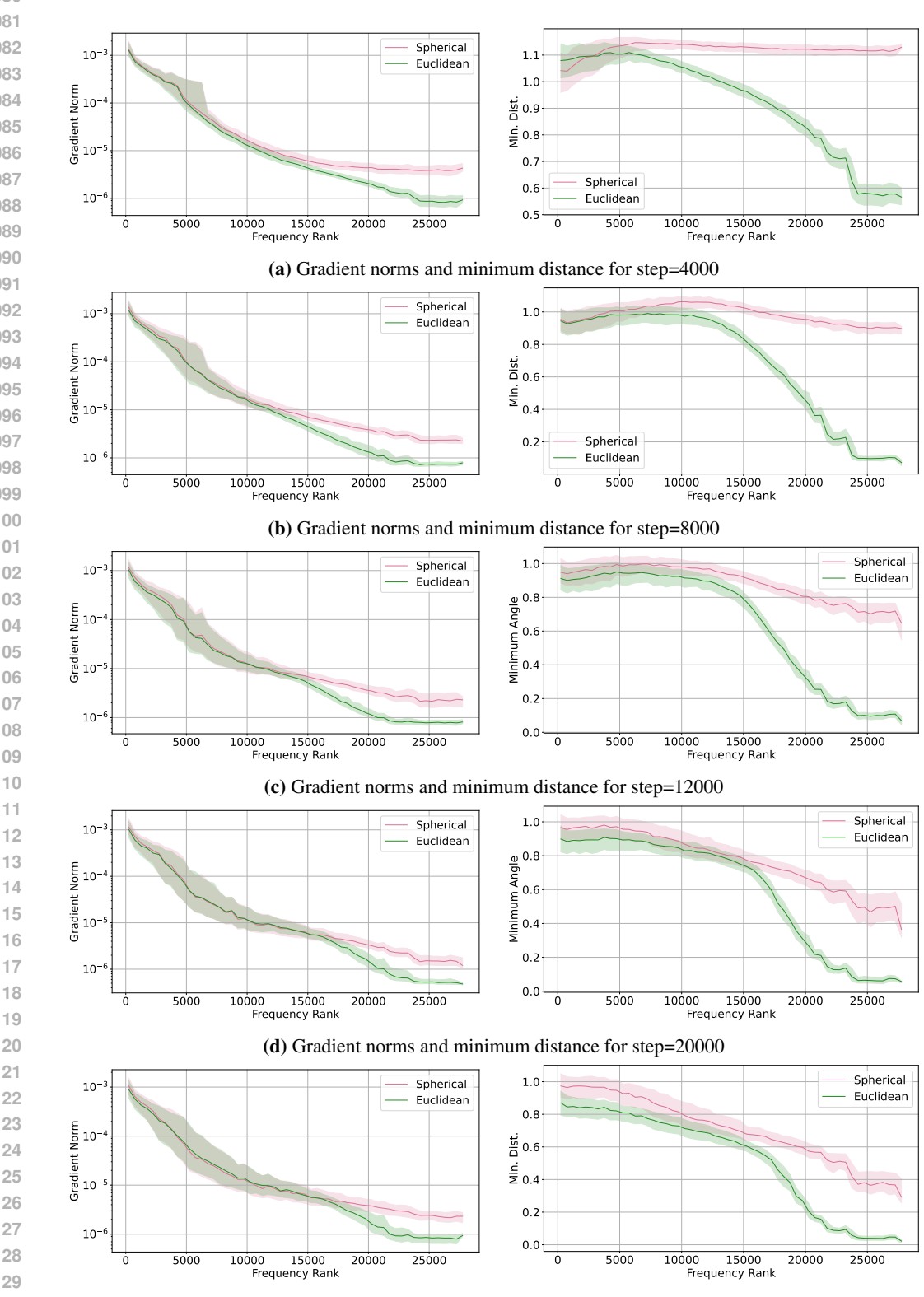

**(a)** Gradient norms and minimum distance for step=4000

**(b)** Gradient norms and minimum distance for step=8000

**(c)** Gradient norms and minimum distance for step=12000

**(d)** Gradient norms and minimum distance for step=20000

**(e)** Gradient norms and minimum distance for step=32000

**Figure 8:** Training dynamic of gradient norms and minimum distances of the target language embeddings.

