# OpenReview forum: "Learning Dispersed Embeddings on Hyperspheres"
_ICLR.cc/2025/Conference — Submitted to ICLR 2025_

### Official Review · Reviewer_tX9P · 2024-11-01

**Soundness:** 3
**Presentation:** 3
**Contribution:** 2
**Rating:** 5
**Confidence:** 3

**Summary:**

The paper proposes methods to improve the spatial separation of high-dimensional embeddings by constraining them to lie on a hypersphere. This separation helps prevent clustering of unrelated embeddings, which can degrade performance in machine learning tasks. The authors reinterpret existing techniques, such as Maximum Mean Discrepancy (MMD) and Lloyd’s algorithm, for dispersing embeddings and propose a novel sliced dispersion method that leverages hyperspherical geometry.

**Strengths:**

1. The paper is well-structured and written in a way that makes it accessible to a broad audience.
2. The background section provides a concise yet thorough summary of relevant literature, including essential mathematical concepts and previous approaches.
3. The experimental evaluation covers a wide range of tasks. The authors evaluate their methods on synthetic data, image classification tasks, and machine translation.

**Weaknesses:**

1. There is little relationship between the background and the proposed methods. It would be beneficial if the authors provided more insights into the proposed sliced dispersion.
2. The training time for empirical datasets is not reported. Given the marginal improvements in image classification and neural machine translation, it would be helpful to understand how the time cost balances against the performance gains.

**Questions:**

1. Correct me if I’m wrong, but by minimizing  L_{sliced} , the authors are attempting to find the optimal great circle  S_{pq}  that aligns with the input  X . How does this improve spatial distribution if the samples are on a lower-rank circle?
2. Visualizing the results from Section 3.1 in 3D space and analyzing how each metric affects the distribution would be valuable.

---

> ### Author Response · Authors · 2024-11-20
> **Reply to reviewer tX9P**
>
> Thank you for your review. Your comments have been helpful in refining our manuscript.
>
> > There is little relationship between the background and the proposed methods. It would be beneficial if the authors provided more insights into the proposed sliced dispersion
>
> Thank you for your comment. We restructure the paper and made a clear distinction between previous work (now Section 2) and our theoretical results (Section 3). We hope it makes it easier to connect the theory with the respective experiments.
>
> > The training time for empirical datasets is not reported. Given the marginal improvements in image classification and neural machine translation, it would be helpful to understand how the time cost balances against the performance gains.
>
> We agree that providing such numbers will further highlight the importance of our proposed methods. Lloyd and Sliced are in general require less memory, since there is no need for pairwise-matrix calculation. We will make sure to add such a comparison in the next revision.
>
> > Correct me if I’m wrong, but by minimizing L_{sliced} , the authors are attempting to find the optimal great circle S_{pq} that aligns with the input X . How does this improve spatial distribution if the samples are on a lower-rank circle?
>
> Thanks for raising this point, we clarified the description of $\mathcal{L}_{sliced}$ in the manuscript. Your description would characterize algorithms like principal geodesic analysis, which seek circles of high variance to best describe the (fixed) data. We seek in a sense the opposite: For dispersion, X is our optimization parameter: we update X in order to maximize dispersion across * all * great circles (in expectation).
>
> The points X are not “moved” to $proj_{Spq}(X)$ when computing eqn (12). As with all algorithms discussed in this paper, each x_i only takes a (likely small) Riemannian step in the direction of its gradient (as given in Prop 1), but the projection onto a great circle is used to determine the direction and magnitude of this update. We added a figure to illustrate this algorithm better in the revised manuscript (Figure 1).
>
> We’re happy to clarify if you have any further questions or concerns
>
> Thank you!

---

> > ### Comment · Reviewer_tX9P · 2024-11-30
> >
> > Thank you to the authors for addressing my concerns. While I appreciate the clarifications provided, I will maintain my original score. Reporting training time would make the paper more convincing, as it would help demonstrate whether the marginal improvements observed in the experiments justify any additional time or memory requirements.

---

### Official Review · Reviewer_FeNY · 2024-11-02

**Soundness:** 2
**Presentation:** 3
**Contribution:** 2
**Rating:** 6
**Confidence:** 2

**Summary:**

The author(s) bridged the gap between the theory of linear separation through the dispersion of embeddings and its application to the high dimensional problem. They introduced a gradient-based method to approximate the optimal dispersion on a hypersphere. They made a systematically comparison of two methods: Lloyd and Sliced.

**Strengths:**

The paper clearly defined dispersion on the hypersphere and provided two methods for the approximation. They compared the methods on various applications under different metrics. They novelly used Riemannian optimization on the hypersphere to achieve dispersion. The improvement on the classification and neural machine translation problem seems promising.

**Weaknesses:**

The paper attempts to optimize the dispersion on the hypersphere but it lacks the justification and discussion why the gradient-based method leads to an optimal dispersion (line 016) in high dimensional space.

**Questions:**

I think the evidence in the numerical experiments supports the improvement made by the dispersion optimization. Could author(s) provide some justifications on how much the dispersion on the hypersphere is related to the improved performance in different tasks? The overall improvement on all of the tasks seems to indicate a great essential improvement on the theory of the method regardless the task (image classification, neural machine translation)?

---

> ### Author Response · Authors · 2024-11-20
> **Reply to reviewer FeNY**
>
> Thank you for your feedback and positive assessment of our work!
>
> We address weaknesses and answer questions below.
>
>
> > The paper attempts to optimize the dispersion on the hypersphere but it lacks the justification and discussion why the gradient-based method leads to an optimal dispersion (line 016) in high dimensional space.
>
> In high-dimensional space, exact optimal solutions for the Tammes problem are unknown. Trade-offs between dispersion and task-related objectives are even harder to solve optimally. Instead, we use numerical methods like Riemannian SGD to approximately optimize the objective, and measure dispersion metrics such as circular variance and minimum distance to quantify the improvements, as well as reporting the task-specific performance metric.
>
>
> > Could author(s) provide some justifications on how much the dispersion on the hypersphere is related to the improved performance in different tasks? The overall improvement on all of the tasks seems to indicate a great essential improvement on the theory of the method regardless the task (image classification, neural machine translation)?
>
>
> Thank you for your great question. There are several works that shows that lack of dispersion harms performance of the model, for example:
> - [The Unreasonable Effectiveness of Random Target Embeddings for Continuous-Output Neural Machine Translation](https://aclanthology.org/2024.naacl-short.56) (Tokarchuk & Niculae, NAACL 2024)
> - [Hubs and Hyperspheres: Reducing Hubness and Improving Transductive Few-shot Learning with Hyperspherical Embeddings.](https://ieeexplore.ieee.org/stamp/stamp.jsp?tp=&arnumber=10204251) (Trosten, et.al., Computer Vision and Pattern Recognition 2023). Based on the previous work and our experiments, we believe reported improvements can be attributed to the dispersion.
>
>
> We would be glad to provide further clarification if you have any additional questions or concerns.
>
> Thank you!

---

### Official Review · Reviewer_aPtT · 2024-11-04

**Soundness:** 3
**Presentation:** 3
**Contribution:** 3
**Rating:** 5
**Confidence:** 3

**Summary:**

The authors investiage representation learning constrained to the sphere, a formulation that is often used in ML approaches.  They introduce some new ways to optimize loss functions and benchmark some approaches.

**Strengths:**

The problem is clearly stated and the motivation is easy to understand.  The paper seems well written and the authors are familiar with theory related work papers.

I find the paper well written and the structure is easy to follow.

Overall I quite like the paper, although I am under the impression that it needs more work

**Weaknesses:**

I appreciate the evaluation, but as it stands it is a bit lacking.  It would be good to show similar related approaches such as SimCLR for the imagenet classification task.  Also, the improvements seem really minor compared to even their own baseline.  On the one hand that is good because supposedly all of the optimizers accomplish the same goal.  On the other hand, it begs the question of why is it useful to use one optimizer over another.  And what is the value proposition of using Riemannian optimization (Bonnabel, 2013; Becigneul & Ganea, 2019)?

There are no repeated runs for the experiments and there is no mention of whether this will holds.  I would like to see some added runs that can then be summarized as mean +- std to give a better idea of how robust the results are.  Especially since the improvements appear quite marginal, this would strengthen the paper in my opinion.

**Questions:**

- The authors cite Wang and Isola (2020) but they make no mention of contrastive learning, which (esp. in Chen et al., 2020) also optimizes an embedding on the hypersphere.  Perhaps the authors wish to comment or include this into their benchmark of ImageNet classification?
- What do the different options for optimization bring to the table?  After reading through the paper this is not fully clear to me.
- Why do you take the «Euclidean» representation learned by a transformer and normalize it to a hypersphere?  Would you not expect this to lead to worse results?
- For Figure 1 and 5, the random initialization seems to always get one of the best results in at least some of the cases.  I find this surprising.  Does it mean that the task can be easily solved with a proper initialization?
- Related to the previous bullet point:  Were the random initializations used for further training?  Because it seems that in Fig. 1 the best metric was actually achieved with a random init.  I would expect the optimizers to not decrease the performance, yet they seem to do so.  Can you explain this?

---

> ### Author Response · Authors · 2024-11-20
> **Reply to reviewer aPtT**
>
> Thank you, we highly appreciate your feedback. We are glad to hear that you liked our paper.
>
>
> We would like to address weaknesses you pointed out in your review and answer questions:
>
>
> > It would be good to show similar related approaches such as SimCLR for the imagenet classification task.
>
> > The authors cite Wang and Isola (2020) but they make no mention of contrastive learning, which (esp. in Chen et al., 2020) also optimizes an embedding on the hypersphere. Perhaps the authors wish to comment or include this into their benchmark of ImageNet classification?
>
> Thank you for your insightful comment! Indeed, contrastive learning serves a similar purpose. The main difference is that in our work, we focus on optimizing parameters rather than optimizing outputs. It is yet to be seen if the dispersion of parameters leads to the dispersion of the outputs. We are currently working in this direction, and we are genuinely interested in studying the connection between contrastive learning and dispersion. We explicitly added the connection to contrastive learning in Section 1 of the revised version.
>
>
> > On the other hand, it begs the question of why is it useful to use one optimizer over another.
>
> > What do the different options for optimization bring to the table? After reading through the paper this is not fully clear to me.
>
> Methods based on the pairwise distances have a main disadvantage: the memory/computational inefficiency, since it requires the calculate of the matrix of all possible pairwise distances. In contrast, Lloyd and Sliced do not rely on pairwise distances and are more memory and computationally efficient. In the revised version, we discuss it under Section 3.
>
>
> > And what is the value proposition of using Riemannian optimization (Bonnabel, 2013; Becigneul & Ganea, 2019)?
>
> That is a great question. We give the detailed answer in the general comment, and revised version contains now discussion on that topic in Section 3.4.
>
>
> > There are no repeated runs for the experiments and there is no mention of whether this will holds. I would like to see some added runs that can then be summarized as mean +- std to give a better idea of how robust the results are. Especially since the improvements appear quite marginal, this would strengthen the paper in my opinion.
>
> Thank you for your suggestion. We are currently working on obtaining more results for each model, and we will definitely include it in the next revision.
>
>
> > For Figure 1 and 5, the random initialization seems to always get one of the best results in at least some of the cases. I find this surprising. Does it mean that the task can be easily solved with a proper initialization?
>
> > Related to the previous bullet point: Were the random initializations used for further training? Because it seems that in Fig. 1 the best metric was actually achieved with a random init. I would expect the optimizers to not decrease the performance, yet they seem to do so. Can you explain this?
>
> Figure 1 and 5 show the minimum angle for each of the points in the set with respect to the dispersion method. Ideal solution is achieved when all angles are on the “optimal angle” line. Random Init represents the angles before the optimization, and indeed optimization is done over these randomly initialized points. We revised Figure 2 (Former Figure 1) and Figure 6 (former Figure 5) to make it more clear.
>
> Thank you!

---

> > ### Comment · Reviewer_aPtT · 2024-11-22
> >
> > I appreciate and acknowledge the authors response.   I will maintain my score as I think it is not yet quite ready (as I think parametric embeddings through e.g. contrastive learning would be beneficial), though I would not mind if the paper was accepted.

---

### Official Review · Reviewer_7g8b · 2024-11-05

**Soundness:** 2
**Presentation:** 3
**Contribution:** 2
**Rating:** 5
**Confidence:** 3

**Summary:**

This paper gives an overview of dispersed embedding. This latter encourages spreading out a large amount of high-dimensional embedding vectors on the surface of the $d$-dimensional unit hypersphere by maximization of the angular angle between any two points. The authors study several dispersion objectives for the nit hypersphere. Then they propose a sliced dispersion version that exploits the geometrical property of the hypersphere.

**Strengths:**

- The paper overviews several measures of embedding dispersion and introduces the Lloyd and Sliced dispersions.
- The authors test the given objectives on synthetic embedding illustrating Tammes problem,  then on image classification with prototypes, and finally on neural machine translation through large-scale text embedding datasets.

**Weaknesses:**

- According to Lemma 2 and Proposition 1, the projection is with respect to two slicing directions $p,q$ on the hypersphere. Empirically, is there any order $L$ of the cardinal of these directions? I mean how many projections are needed to guarantee maximum data dispersion?
- Does the target dimension $d$ belong to the set of hyperparameters
- The optimization algorithm is structured with respect to Riemannian optimization (RO) tools. Can you highlight the benefits of RO in the learning process?
- The paper is fully empirical which is certainly important to showcase the benefits of sliced dispersion over the SOTA approaches. However,  I think it lacks some theoretical insights in comparison with respect to euclidean baselines.

**Questions:**

See Weakness section.

---

> ### Author Response · Authors · 2024-11-20
> **Response to reviewer 7g8b**
>
> Thank you very much for your feedback. Below are our responses to your questions.
>
> > According to Lemma 2 and Proposition 1, the projection is with respect to two slicing directions
>  on the hypersphere. Empirically, is there any order
>  of the cardinal of these directions? I mean how many projections are needed to guarantee maximum data dispersion?
>
> Thank you for your great question! There are infinitely many great circles even in 3d. Our optimization objective is defined to take all of them into account via the expectation, which we optimize with stochastic methods. As common in Monte Carlo methods for gradient estimation, we don't take multiple samples at once, but rather a small number of samples (in our case 1 sample) for each gradient update, but perform sufficient gradient updates. We provide the plot in Appendix A.3.
>
> > Does the target dimension belong to the set of hyperparameters
>
> Yes, that is a hyperparameter. We have conducted experiments with different numbers of dimensions, and in all cases, they showed similar trends. We have chosen 128 since it is sufficiently large but still computationally feasible.
>
> > The optimization algorithm is structured with respect to Riemannian optimization (RO) tools. Can you highlight the benefits of RO in the learning process?
>
> This is a good question. We agree that it was unclear in our submission. We address it in a general comment, and also added in revised version Section 3.4 dedicated to the motivation of using RO.
>
> > The paper is fully empirical which is certainly important to showcase the benefits of sliced dispersion over the SOTA approaches. However, I think it lacks some theoretical insights in comparison with respect to euclidean baselines.
>
> We would appreciate it if you could clarify this question. In the revised section 3, we provide more insights into the Sliced dispersion and also give a theoretical justification for using RO. If we can provide more details on anything in particular to better answer your question, we will be happy to do so.
>
> Thank you!

---

> > ### Comment · Reviewer_7g8b · 2024-11-26
> >
> > I thank the author for their answers to my concerns. I will keep my score unchanged.

---

### Official Review · Reviewer_WTYB · 2024-11-09

**Soundness:** 3
**Presentation:** 2
**Contribution:** 2
**Rating:** 5
**Confidence:** 2

**Summary:**

This paper compares different methods to compute dispersed embeddings in the hypersphere. This amounts to solving the "Tammes problem" in the hypersphere which is relevant to many applications. It first introduces a panorama of the different existing approaches  before benchmarking them on various tasks.

**Strengths:**

- Computing dispersed embeddings on the hypersphere is a highly relevant problem for the representation learning community.
- The review presented at the beginning of the paper is informative.
- The authors explore a variety of settings in their experiments.

**Weaknesses:**

- Overall, I find the paper somewhat disorganized, which makes it challenging to identify the authors' unique contributions. These new insights should be more clearly highlighted in contrast to previous work.

- Section 2 feels more like a catalog of methods without adequate comparison. There isn't a clear progression that clarifies the rationale or direction. For example, why is max-min considered superior to MMD? Additionally, there should be an evident relationship between Differential Entropy Dispersion and MMD projection with uniform distribution. Emphasizing this connection would add valuable insight.

- The clarity of this paper could be significantly improved by stating the objective of each subsection at the beginning and explaining its relevance within the overall context of the paper.

- While the experimental section is diverse, it’s challenging to understand which method the authors recommend. Providing more practical guidelines for practitioners on what works best would be helpful. For example, why does the sliced method perform well in Neural Machine Translation (NMT) but not in prototype-based image classification?

**Questions:**

- Could the authors elaborate on the motivation for using Riemannian optimization over projections?

- Did the authors consider conducting experiments with self-supervised computer vision models, where representation learning on the hypersphere is common?

- Typo: "dipsersion" should be corrected to "dispersion" on line 372.

---

> ### Author Response · Authors · 2024-11-20
> **Response to reviewer WTYB**
>
> Thank you so much for your review, it helped us to improve the paper greatly. We are also happy to hear that you think the stated problem is highly relevant for the community. Below we address your main concerns and answer question:
>
>
> > Overall, I find the paper somewhat disorganized, which makes it challenging to identify the authors' unique contributions. These new insights should be more clearly highlighted in contrast to previous work.
>
> > Section 2 feels more like a catalog of methods without adequate comparison. There isn't a clear progression that clarifies the rationale or direction. For example, why is max-min considered superior to MMD?
>
> > stating the objective of each subsection at the beginning and explaining its relevance within the overall context of the paper.
>
> Thank you so much for your valuable suggestions! We agree that clear distinction of our contributions is important, so we revised the manuscript accordingly. Specifically, we split section two into two separate section: **Section 2** now focuses on background and existing dispersion methods from the literature, **Section 3** emphasise the dispersion optimization methods we propose. The order of describing dispersion methods is arbitrary, and does not induce the superiority or inferiority of particular method. We also added brief description in the beginning of each section.
>
>
> > Additionally, there should be an evident relationship between Differential Entropy Dispersion and MMD projection with uniform distribution. Emphasizing this connection would add valuable insight.
>
> That is a very good point, and we included the brief description in the revised version. We draw a connection through uniformity loss defined in the Section 2.3, and MMD is the re-interpretation of uniformity loss.
>
> > Providing more practical guidelines for practitioners on what works best would be helpful. For example, why does the sliced method perform well in Neural Machine Translation (NMT) but not in prototype-based image classification?
>
> We agree that adding practical guidelines will be valuable, thank you for bringing this up. Our general recommendation is to use sliced when $d \ll N$, since computing pairwise-based metrics is expensive and often not possible, and we must rely on batched version (consider only part of the matrix for calculation), in our experience sliced performs well when $d \ll N$. In other cases MMD is a good candidate for optimizaing dispersion. It is also a trade-off between alignment and dispersion, since larger dispersion in some cases can hard the model (e.g., as shown in section 4.3). We will add more details in the next revision.
>
> >Could the authors elaborate on the motivation for using Riemannian optimization over projections?
>
> We admit that motivation for using Riemannian optimization was unclear, thank you for pointing this out. We address this question in general comment, and in the revised version we added Section 3.4 that discusses the motivation for choosing RO.
>
> >Did the authors consider conducting experiments with self-supervised computer vision models, where representation learning on the hypersphere is common?
> We indeed consider including self-supervised computer vision models. We have done some preliminary experiments on a small model and dataset with 10 classes (STL10). However, it is hard to observe the impact of dispersion on such a small scale. We want to conduct experiments on a larger scale and include them in the next revision.
>
> We are happy to elaborate more on any of your questions if necessary. Thank you!

---

### Author Response · Authors · 2024-11-20
**General comment: Thank you to all reviewers & summary of the revisions**

We thank all reviewers for their valuable feedback, which has been instrumental in improving our submission. We have now uploaded a revised version of the manuscript. All the changes are highlighted in a different color. Below is a summary of the updates:

- In **Section 1**, we explicitly discuss the connection to contrastive learning methods.
- The paper has been restructured: the original Section 2 has been split into two sections. **Section 2** now provides an overview of dispersion objectives known in the literature and relevant background information, while **Section 3** focuses on our contributions.
- In **Section 3.3**, we added **Figure 1**, which visualizes one step of sliced dispersion and highlights the differences compared to geodesic PCA.
- To address the reasoning behind using Riemannian optimization, we added a detailed justification in **Section 3.4**.
- We corrected typos, added a missing caption, and included brief descriptions for each section.

Reviewers WTYB, 7g8b05, and aPtT raised concerns about the necessity of Riemannian optimization. We acknowledge that this aspect was not adequately addressed in the initial submission. In the revised version, we provide a theoretical justification for using Riemannian optimization as an alternative to constrained optimization in $\mathbb{R}^d$ in **Section 3.4**. We also include an empirical comparison in **Appendix B**. Below is an excerpt from the revised version:

> Optimization for dispersion can be defined as constrained optimization problem in $\mathbb{R}^{d}$, where constraint is that points lie on the hypersphere. This can be solved by ignoring spherical constrains and projecting the parameters onto the sphere after the gradient update, however convergence is not guaranteed, because the sphere is not a convex set, even though it can give acceptable results with careful initialization (Raman & Yang, 2019). Alternatively, we can rely on Riemannian optimization (Bonnabel, 2013; Becigneul & Ganea, 2019) in $\mathbb{S}^{d−1}$ as effective unconstrained extension (Bloch,2015; Boumal, 2023) with guaranteed convergence (Bonnabel,2013). We further empirically explore the convergence of both methods in Appendix B.


We are happy to address any further questions or concerns.

---

### Meta-Review · Area_Chair_KvaG · 2024-12-22

**Metareview:**

The authors propose dispersion learning for embeddings on the hypersphere. The authors proposed sliced dispersion and leverage Riemannian optimization to optimize the dispersion for the embeddings. The Reviewers raised concerns on experiments, e.g., why embedding Euclidean data from transformer into the hypersphere, the empirical benefit for optimizing the dispersion for embeddings on the hypersphere is not well-supported. It is also better to motivate the advantages of Riemannian optimization and its empirical benefits for the proposed approach. Given aforementioned concerns, it leads to a lack of enthusiastic supports from the Reviewers. The authors may consider the Reviewers' comments to improve the submission.

**Additional Comments On Reviewer Discussion:**

The Reviewers raised several points as mentioned in the meta-review on the experimental setting, motivation, advantages and empirical supportive evidence of using Riemannian optimization and dispersion objective in embeddings on the hypersphere. Thus, we think the submission is not ready for publication yet.

---

### Decision · Program_Chairs · 2025-01-22

Reject